

# Classification accuracy and compatibility across devices of a new Rapid-E+ flow cytometer

Branko Sikoparija[1], Predrag Matavulj[2], Isidora Simovic[1], Predrag Radisic[1], Sanja Brdar[1], Vladan Minic[1], Danijela Tesendic[3], Evgeny Kadantsev[4], Julia Palamarchuk[4] and Mikhail Sofiev[4]

[1]BioSense Institute Research Institute for Information Technologies in Biosystems, University of Novi Sad, Novi Sad, 2100, Serbia
[2]Institute for Data Science, University of Applied Sciences North Western Switzerland, Windish, 5210, Switzerland
[3]Department of mathematics and informatics, Faculty of Sciences, University of Novi Sad, Novi Sad, 21000, Serbia
[4]Finnish Meteorological Institute, Helsinki, Erik Palmenin Aukio 1, FI-00560, Finland

*Correspondence to*: Mikhail Sofiev (Mikhail.Sofiev@fmi.fi)

**Abstract.** The study evaluated a new model of a Plair SA air flow cytometer, Rapid-E+, and assessed its suitability for airborne pollen monitoring within operational networks. Key features of the new model are compared with the previous one, Rapid-E. A machine learning algorithm is constructed and evaluated for (i) classification of reference pollen types in laboratory conditions and (ii) monitoring in real-life field campaigns. The second goal of the study was to evaluate the device usability in forthcoming monitoring networks, which would require similarity and reproducibility of the measurement signal across devices. We employed three devices and analysed (dis-)similarities of their measurements in laboratory conditions. The lab evaluation showed similar recognition performance as that of Rapid-E, but field measurements in conditions when several pollen types are present in the air simultaneously, showed a notably lower agreement of Rapid-E+ with manual Hirst-type observations than those of the older model. An exception was the total-pollen measurements. Comparison across the Rapid-E+ devices revealed noticeable differences in fluorescence measurements between the three devices tested. As a result, application of the recognition algorithm trained on the data of one device to another one led to large errors. The study confirmed the potential of the fluorescence measurements for discrimination between different pollen classes, but each monitor needed to be trained individually to achieve acceptable skills. A large uncertainty of fluorescence measurements and their variability between different devices need to be addressed to improve the device usability.

## 1 Introduction

A recently published special issue "Bioaerosol Research: Methods, Challenges, and Perspectives" provided an extensive overview of developments in monitoring of primary biological aerosol particles, emphasizing the interest in real-time automatic measurements (Huffman et al., 2019). In the past 10 years, several devices were released to the market claiming to be able to detect and quantify atmospheric concentrations of various bioaerosols (i.e. pollen and fungal spores) (Butters et al., 2022). An extensive international intercomparison of automatic bioaerosol monitors with reference measurements (EN 16968, 2019) was organised within the framework of the EUMETNET AutoPollen Programme and the ADOPT COST Action in



2021. It indicated that three automatic instruments, with an appropriate identification algorithm, are capable of identification of the main types of airborne pollen present in the atmosphere of Munich during the campaign: Hund BAA-500, Swisens Poleno (Mars and Jupiter models), and Plair Rapid-E (Maya-Manzano et al., 2023). They also showed high reliability, which

made them potentially suitable for a continuous pollen monitoring within operational networks of automatic aerobiological stations. The campaign has also raised some concerns regarding the device calibration and inter-calibration, to be addressed in follow-up studies and campaigns.

The aim of this study is to evaluate a new model of Plair air flow cytometers, Rapid-E+, and assess its suitability for operational automatic measurements of airborne pollen and fungal spores within forthcoming monitoring networks. We

performed a series of laboratory experiments and evaluated the device performance in real-life field conditions by comparing their measurements with the standard manual method. In addition to testing the recognition performance of certain bioaerosols, we have analysed to what extent different devices are compatible with each other and thus allow for a common classification algorithm trained with data collected with one (or a few) device(s) and applied across the network.

## 2 Material and methods

### 2.1 Rapid-E+ flow cytometer at a glance: pros- and cons- of the new model

In this study, we are focussing on the air flow cytometer Rapid-E+ from Plair SA (http://www.plair.ch), which is a new model stemming from PA-300 (Crouzy et al., 2016) and Rapid-E (Sauliene et al., 2019). Although the same approach for measuring particle morphology (laser scattering) and chemical characteristics (laser-induced fluorescence spectrum and lifetime) is used, Rapid E+ substantially differs from its predecessor (Table A1). In particular, Rapid-E+ samples 5 l min$^{-1}$, and records all

particles passing through a 447 nm scattering laser into 4 size bins (>0.3 µm, >0.5 µm, >1 µm, >5 µm). High efficiency of detections has been verified for the device prototype at Swiss Federal Institute of Metrology (Certificate of Calibration No. 235-11067): >80 % of particles ranging within 0.5-5 µm and 65-75 % of particles of a 5-10 µm range. Unfortunately, a test for larger particles (in a range of most pollen grains) has not been performed. Like in its predecessor, the fluorescence measurements of Rapid-E+ can be limited to particles within a specific size range (i.e. 0.3-100 µm, 1-100 µm, 5-100 µm), thus

ignoring smaller and larger particles, to extend the excitation-inducing laser lifetime. Changing the particle size sensitivity also allows adapting the gain of detectors of the fluorescence spectrum and lifetime, which is useful for measuring particles with low fluorescence emission, such as most fungal spores. The lifetime of the 337 nm laser has been extended, according to the manufacturer, from about 100 million to about 200 million shots. However, recording all particles larger than 1 µm could easily result in 2000 particles per minute measured, which would still quickly use up the laser. The device offers a solution by

enabling intermittent high sensitivity measurements (e.g. one in every ten minutes).

Each measurement component in Rapid-E+ went through changes compared to its predecessor: the 447 nm laser scattering is measured now in two polarization planes at a narrower angle window, the fluorescence spectrum and the fast speed fluorescence decay (lifetime) are measured at a narrower wavelengths range. The device also records slow speed fluorescence



decay by measuring spectrum at the moment of the 337 nm laser shot and then followed by 31 measurements every

microsecond. In addition, the scattering image from a 637 nm laser is recorded with a 4x4 pixel detector.

Interface of the device has been changed as well and generally became less convenient. Rapid-E+ output files contain data of 10000 particles each and there is no more time stamp in the file name. In addition, the data transfer protocol from the device storage changed from SSH of Rapid-E to SFTP, which has limitations in handling security keys, so the remote file synchronisation (rsync) is not supported anymore. It complicated the automatization of the data download to an external storage

and forced a reprogramming of the external operational environment after the upgrade from Rapid-E.

## 2.2 Experiments with Rapid-E+

Three Rapid-E+ air flow cytometers have been involved in this study. One device operated in Novi Sad (serial number 00E7277C) was trained in theNovi Sad  laboratory and then set into continuous field measurements during period 7 April – 27 September 2023. Two other devices, owned by City of Osijek in Croatia (serial number 00E74EDE) and Finnish

Meteorological Institute (FMI) in Helsinki (serial number 00C59ACA), were used in the corresponding laboratories to test compatibility of the devices and transferability of the pollen recognition algorithm.

### 2.2.1 Field monitoring campaign

The monitoring was performed at a roof level (20 m a.g.l.) in Novi Sad (45.245575° N, 19.853453° E). The test period allowed to explore measurement performance of automatic bioaerosol monitor in a variety of conditions characteristic for Pannonian

plain i.e. a large diversity of pollen (Tesendic et al., 2020) and fungal spores (Simovic et al., 2023) often mixed with abundant mineral dust (Sikoparija et al., 2020), but also occasional episodes of unusual bioaerosols, such as starch (Sikoparija et al., 2022). In the study region, the period of seasonal allergies is extended by the weed pollen season from July to the end of October when large quantity of ragweed pollen is recorded in the air (Sikoparija et al., 2018).

During the campaign, the sensitive "Middle mode" (all particles coarser than 1 μm) was active for one minute in ten minutes

cycles, which resulted in six equidistantly one-minute measurements per hour, which is still representative for capturing tmain features of the diurnal variations, albeit at a somewhat coarser temporal resolution (Sikoparija et al., 2020).

### 2.2.2 Laboratory measurements of bioaerosols

Laboratory work aimed at two goals:

(i)    we created an extensive training dataset using the device operated in Novi Sad. Reference pollen for training has been

90         collected locally. We selected 27 pollen classes (Table A2) that represent the most abundant pollen in Novi Sad. To explore the specificity of chemical analysis from fluorescence measurements, the selected classes include pollen classes that are morphologically similar (e.g. Cannabis and Humulus, Juniperus and Taxus, Urtica and Parietaria), which are commonly grouped together in manual identification. The laboratory tests were performed in two different sensitivity modes: "Pollen mode" that measures fluorescence for particles larger than 5 μm and "Middle mode" that measures



95    fluorescence for particles larger than 1 µm with 10% increased sensitivity of the lifetime detector and 28% increased sensitivity of the spectrometer.

(ii)  Two other devices were tested independently in Osijek and Helsinki, respectively, with subsets of the Novi Sad pollen collection in order to produce theoretically-identical training datasets for the corresponding pollen types. To ensure identity, the pollen samples from Novi Sad were shared between the laboratories. Training of both devices was

100    performed in the "pollen mode".

### 2.2.3 Reference data collection

The FMI Rapid-E+ device was exposed to pollen by using Swisense Atomizer (Swisens, 2023). A custom-made system (Bruffaerts et al., in preparation) with similar features was developed to expose Novi Sad and Osijek devices. Both systems prevent particles from the ambient air from entering the detection chamber while keeping the sampling flow unaffected and

105 facilitating the emission of pollen from an Eppendorf cuvette by a combination of vibrations and air blows. The devices were exposed to pollen until a sufficient number of particles was collected for training, validatiing, and testing a classification algorithm (Table A2). Since the atmosphere also contains numerous aerosols other than pollen (e.g. fungal spores, mineral dust, starch), an additional training class was created from operational measurements containing particles measured at the roof during periods when no pollen was recorded in samples collected in a side-by-side operating Hirst type sampler.

110 The data were pre-processed prior to further analysis (Figure B1). Firstly, we removed measurements at the seventh and the eighth bands of the fluorescence spectrum, which, according to the manufacturer, record light at about 450 nm and at about 462 nm, respectively, thus being affected by the scattering laser interference. Only five spectral measurements (i.e. fourteenth-eighteenth acquisitions corresponding to 13-17 µs from laser triggering) were used for classification of bioaerosols. Each spectrum measurement, as well as both scattering images, were smoothed with the Savitzky-Golay filter (Savitzky and Golay,

115 1964) for the noise removal. Lifetime of fluorescence measurements was aligned to start at the 4th pixel before the first maximum to avoid shifts caused by temperature changes in the device. Both the fluorescence spectrum and the fluorescence lifetime modalities were converted into image-like formats for further neural network processing and then normalized into 0-1 range to focus on the shape of the signal rather than its intensity. This resulted in the following input data dimensions: 14 x 5 for the fluorescence spectrum, 22 x 3 for the fluorescence lifetime, 120 x 14 for both polarization scattering, and 4 x 4 for

120 "infrared" scattering, as illustrated in Figure B1.

The data were also filtered to remove particles for which noise exceeded the signal. To do this, we focused on intensity of the scattering and fluorescence signals, as it was done in previous studies with Rapid-E (Tesendic et al., 2020; Matavulj et al., 2022; 2023; Sikoparija et al., 2022; Brdar et al., 2023). The particles, for which the maximum intensity of the spectrum did not exceed 4000 units or a sum of scattering measurements was below 50000 units after smoothing, were removed from the

125 analysis (Table B2). The class "other" included 1942375 particles, out of which only 10282 remained after filtering. In the more sensitive "middle mode", 54776 out of 1156902 particles remained in the class "other" after filtering. The single particle



measurements showed very large variability even within filtered dataset (Figure B2), seemingly larger than in the case of Rapid-E (Sauliene et al., 2019).

### 2.2.4 Creating classification algorithm

In the current study, we applied a two-step classification. The first step separates pollen from the class "other", whereas the second step classifies particles recognised as pollen at the first step into 27 pollen classes. The AI-based classification model combined all measurement modalities (i.e. parallel polarization scattering, perpendicular polarization scattering, infrared scattering, fluorescence spectrum, and fluorescence lifetime), assuming that it will result in the best performance as it was the case for Rapid-E (Tesendic et al., 2020).

The ResNet architecture with shortcut connections was employed due to its demonstrated superior performance in classifying pollen based on Rapid-E measurements (Matavulj et al., 2023, Daunys et al, 2022). Due to variability of input data, a variation of 18-layer ResNet model was implemented (i.e. a 4-block-layer for the fluorescence spectrum and lifetime, a 3-block-layer for the 447 nm laser scattering images, and a 1-block-layer for the 672 nm laser scattering image). The block-layers contained either three convolutional layers or four in a case of the first block-layer. In each block layer, we captured a residual following

the initial convolution. Subsequently, at the closure of each block layer, we established a residual connection to the layer's output. Following the completion of all block layers, an additional convolutional layer was integrated. This was followed by a global average pooling, which averaged over the spatial dimensions of the images. The network initially learned from each type of input separately. After this initial training, we transferred the learned features from these individual inputs (specifically, the parts of the network responsible for feature extraction, known as convolutional blocks) to a new network. This new network

processed all different inputs together by equalizing the features from each input using a fully connected (FC) layer, which were then merged. Finally, the network was trained only to classify this combined data using another FC layer with a SoftMax function. During this phase, the weights of the feature extractors (the convolutional blocks) were kept unchanged. This means that while the network was learning to classify the merged data, the initial parts that extract features from each input type did not undergo any further changes.

The first convolutional layer was customized to accept a monochrome image. For handling the lifetime and spectrum data, this layer was configured with a kernel size of 5x5, a padding of 2x2, and without any stride to maintain the original spatial dimensions. The classification model was trained with 80 % of the reference dataset, 10 % of particles were used for model validation during training to avoid overfitting, and 10 % were used to test the classification performance after the training.

### 2.3 Manual measurements of bioaerosols in the field campaign

The performance of Rapid-E+ in the field bioaerosol monitoring has been assessed by comparing its 2-hour averaged pollen concentrations with values obtained from the Hirst-type manual standard method EN16868 (CEN, 2019), following the approach described by Matavulj et al. (2022).





Lanzoni VPPS2000 volumetric pollen and spore trap of the Hirst (1952) design situated side-by-side to Rapid-E+ continuously sampled the ambient air at 10 l min $^{-1}$ through a 2 mm×14 mm orifice constantly oriented towards the direction of the wind.

Particles sampled with the airflow were impacted onto an adhesive transparent plastic tape that was mounted on a rotating drum moving past the orifice at 2 mm h$^{-1}$. The 48 mm long tape segments corresponding to 24-h periods were subsequently mounted onto a microscope slide and analysed by a light microscope at ×400 magnification. Pollen grains were counted along three horizontal transects corresponding to 11.57 % of the slide following EN16868 requirement (CEN, 2019), while fungal spores were counted along one horizontal transect (i.e. 3.86 % of the sample) following the recommendation of Galan et al.

(2021). The results were expressed as pollen m$^{-3}$ (Galan et al., 2017).

### 2.4 Meteorological data

Meteorological measurements were obtained from an automatic meteorological station (INOVIS15, Dinarska 2, 21000 Novi Sad, 45.236° N, 19.809° E) located about 3.5 km from the aerosol measurements. The data for relative humidity, wind speed, and precipitation were retrieved from a Weather Underground database

(https://www.wunderground.com/dashboard/pws/INOVIS15).

### 2.5 Data analysis

Agreement between the automatic and the standard manual measurements was quantified via temporal correlation coefficient. The correlation was evaluated for daily pollen concentrations to limit the shot-noise uncertainty resulting from substantial detection limits due to the limited flow rate of the devices (Tummon et al., 2022). The correlations were calculated both for

the entire measurement period (to account for the effect of false positives outside the main flowering season) and for days when average pollen concentrations measured by the manual method exceeded 10 pollen m$^{-3}$, a suggested threshold for calculating the uncertainty by the standard EN 16868:2019. By following this approach, we also focused on the main pollen season thus limiting the inflation of correlation coefficients and p-values due to seasonality. Initial data assessment using the Shapiro-Wilk test was performed to check for normality of distribution. Where data were found to be normally distributed,

Pearson correlation analysis was applied; Spearman's correlation coefficient was calculated otherwise.

## 3 Results and discussion

### 3.1 Aerosol quantification

The Rapid-E+ measurements in Novi Sad had only four interruptions (from 12 to 36 hours long) during six months of the continuous operations. One resulted from a physical blockage of the nozzle, which was resolved by cleaning. The other three

resulted from a software "bug" related to flow measurements, which was switching off the 337 nm laser. Those cases were resolved by restarting the device, which had to be done manually on the roof.

A strong feature of the device was its ability to provide output with very high temporal resolution (Fig.1).



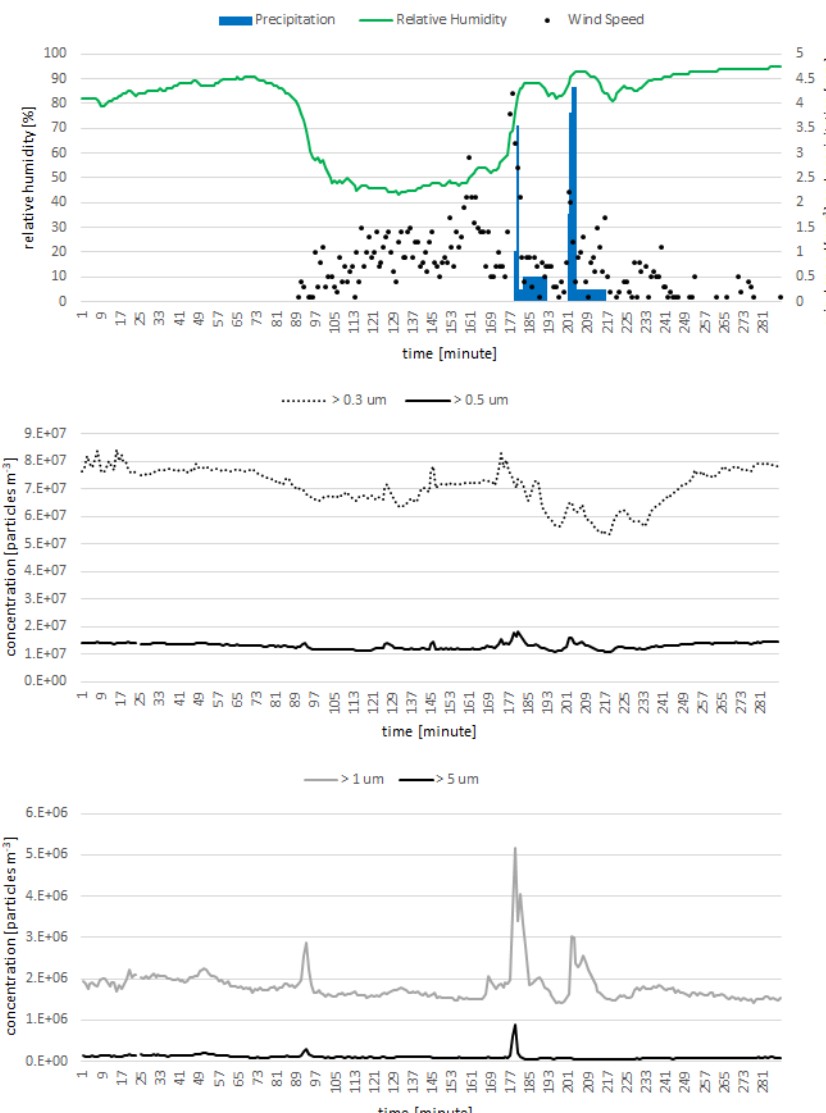

**Figure 1: Timeseries of 5-minute average meteorology measurements i.e. relative humidity, wind speed, precipitation (upper panel) and 5-minute average fine particle concentrations (middle panel) and coarse particle concentrations (lower panel) measured by Rapid-E+ bioaerosol monitor. The time axis shows minutes from 00 UTC on 30 July 2023.**

The concentrations of submicron particles were notably higher that those larger than 1 µm and 5 µm. We also registered several sharp increases of the detected particles, seemingly related to approaching atmospheric fronts and rain episodes (Fig. 1). It was the most pronounced for particles larger than 1 µm (Figure 1). It is interesting to note that after the rain start the coarse particles (> 5 µm) did not follow the increase in concentrations of small aerosols. This observation emphasizes the advantage of measuring with a high temporal resolution simultaneously resolving particle size distribution, for exploring the behaviour of aerosols in changing meteorological conditions. However, quite low flow rate (5 l min$^{-1}$) limited the temporal resolution of the observations (Tummon et al., 2022).



## 3.2 Pollen recognition performance in laboratory

Performance of the binary model designed to discriminate pollen from "other" bioaerosols measured in "pollen mode" (Fig.
C1A) in laboratory conditions was characterised by high precision (94 %), recall (98 %) and F1 score (0.96). Classifications
of twenty-seven pollen classes in "pollen mode" (Fig. 2A) yielded precision, recall and F1 score at 83 %, 85 %, and 0.84,
respectively, which was comparable to results of classification models built for the Rapid-E measurements for the similar
number of pollen types (Tesendic et al. 2020, Smith et al. 2022, Matavulj et al. 2022). As expected, there was a confusion
between Alnus, Betula and Corylus, Morus and Broussonetia, Carpinus and Quercus and Alnus, Cannabis and Humulus and
Morus, which have similar morphology. Once we merged those classes that cannot be distinguished in the manual analysis
(i.e. Cannabis and Humulus, Juniperus and Taxus, Urtica and Parietaria) the performance improved (precision, recall and F1
score are 86 %, recall 86 % and F1 score 0.86). It is interesting to note that classification algorithm with high accuracy
distinguished Urtica and Parietaria from Brousonetia despite these pollens are morphologically similar. However, there was
an unexpected confusion between Cannabis and Platanus.
Measurements with the more sensitive "middle mode" resulted in more particles exceeding the fluorescence threshold (Table
A2). However, as can be seen from confusion matrix (Fig. 2B) the performance in discriminating pollen from other aerosols
slightly decreased. The precision, recall and F1 score were 93 %, recall 96 % and F1 score 0.95, respectively (Fig. C1B).
Performance of the multiclass pollen classification also decreased, so that precision, recall and F1 score became 75 %, 77 %,
and 0.76, respectively. The accuracy improved only for Corylus.




(A)    (B)

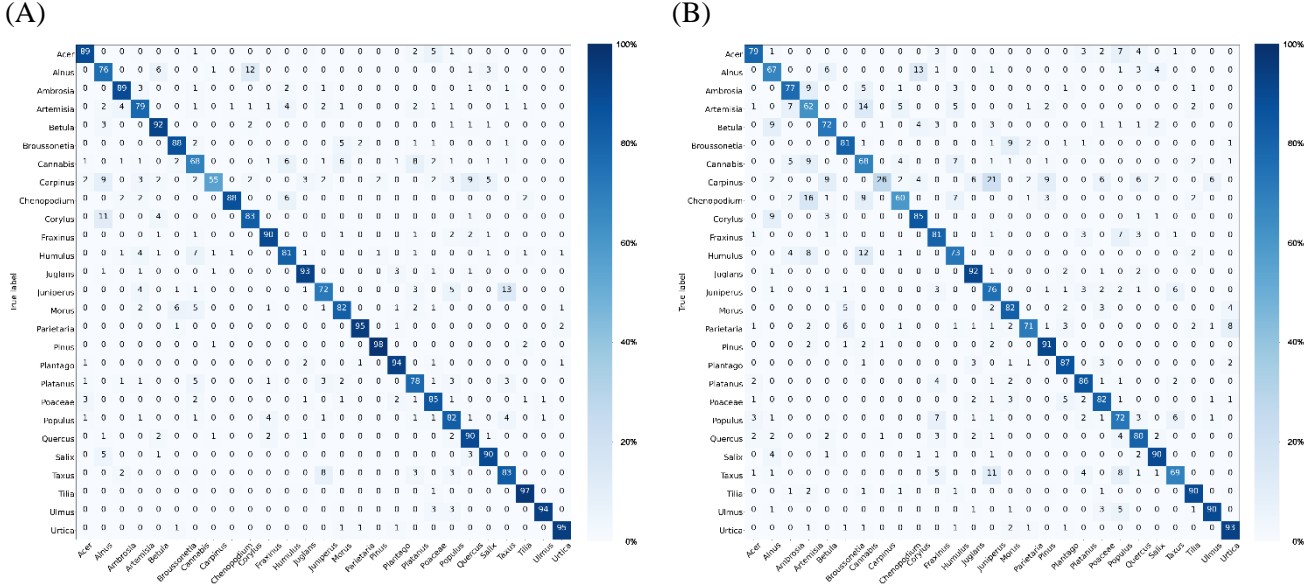

**Figure 2: Confusion matrices depicting pollen classification performance on test dataset measured in (A) "pollen mode" and (B) "middle mode".**


## 3.3 Comparison of field measurement with manual reference time series

The Rapid-E+ measurements in "pollen mode" record an order of magnitude less pollen and fungal spores than the Hirst-type measurements (Fig. 3). This can be attributed to the very rigorous cleaning of the measurements (as described in Section 2.2.3), either from failed measurements (in particular, fluorescence) or good measurements of particles that emit a weak fluorescence

signal.

From the 27th to the 31st daily measurement points, Rapid-E+ underestimated total pollen concentrations even more. When looking into the pollen types detected by the standard measurements for these days, a notable amount of small Broussonetia pollen (about 10 µm (Halbritter 1998)) is evident (Fig. C2), which probably caused the higher omission rate. The apparent under-representativity of the Rapid-E+ measurements for small pollen grains could be handled by a less strict cleaning of the

scattering signal. This would improve detections of Broussonetia, Urtica, Morus, Parietaria, Platanus but could increase the number of false positives from other small aerosols present in the atmosphere. Similar underestimation can be seen for the days 136 - 144 (corresponding to 21-29 August) when Ambrosia pollen was dominant in the atmosphere, implying that notable amount of this pollen was also filtered out. Ambrosia has larger diameter but contains air in its pollen wall (like saccate pollen i.e. Pinus, Picea, Abies), which could affect refraction index and resulted in a size underestimation when inferred from more

homogenous PSLs. Also, it could affect the fluorescence measurements by limiting the number of excited fluorophores, which in turn would require more sensitive detections of fluorescence for reliable counting.



(A)

(B)

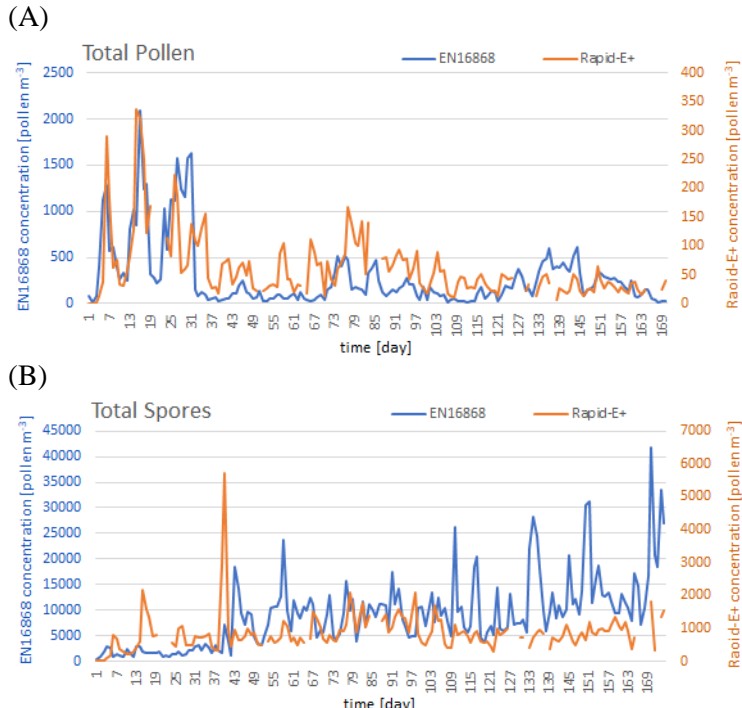

**Figure 3: Time series of daily concentrations measured in Novi Sad by side-by-side operated Hirst type device (EN16868) and Rapid-E+ bioaerosol monitor in "pollen mode" for (A) total pollen and (B) total spores (mind the difference in y-axes scale).**


Automatic detections of total pollen, so as Juglans, Morus and Ambrosia, are in statistically significant positive correlation with the standard EN16868 measurements during days when daily concentration exceeded 10 pollen m$^{-3}$ (Table 1). Overall seasonality was captured for most of pollen classes with a limited number of false positive detections outside the season. The exceptions were Juglans, Pinus, Tilia, Chenopodium, Humulus, and Cannabis, for which a significant number of wrong
classifications existed outside the pollen season (Fig. C3). There was a clear tendency towards confusion of pollens occurring simultaneously in the air, which was expected following the result of tests shown in Fig. 2A.

**Table 1: Correlations between daily concentrations measured by Rapid-E+ in "pollen mode" and EN16868 measurements.**

|  | All days | Concentration > 10 pollen m$^{-3}$ (number of data points in bracket) |
|---|---|---|
| Total Pollen | 0.378 ** | 0.583 [a] (153) ** |
| Total Fungal Spores | 0.060 | 0.180 [a] (156) * |
| Acer | 0.117 | - |
| Alnus | 0.237 ** | - |
| Ambrosia | 0.642 ** | 0.693 [a] ** (41) |
| Artemisia | 0.342 ** | - |





| Betula | 0.680 ** | 0.795 ** (16) |
|---|---|---|
| Broussonetia | 0.703** | 0.386 [a] (21) |
| Cannabaceae | Cannabis 0.082, Humulus 0.477 ** | Cannabis -0.721 (6) Humulus -0.540 [a] (6) |
| Carpinus | 0.557 ** | - |
| Chenopodium | 0.626 ** | 0.534 [a] (6) |
| Corylus | -0.103 | - |
| Fraxinus | 0.496 ** | 0.345 [a] (4) |
| Juglans | 0.180 * | 0.345 [a] (19) |
| Morus | 0.744 ** | 0.576 [a] **(25) ** |
| Pinaceae | 0.187 * | 0.186 [a] (13) |
| Plantago | 0.137 | 0.338 [a] (15) |
| Platanus | 0.659 ** | 0.766 ** (16) |
| Poaceae | 0.454 ** | -0.110 [a] (58) |
| Quercus | 0.633 ** | 0.317 [a] (20) |
| Salix | 0.652 ** | 0.582 [a] *(19) |
| Taxaceae/Cupressaceae | Taxus 0.549 **, Juniperus 0.462 ** | Taxus -0.632 [a] (3), Juniperus -0.900 [a] (3) |
| Tilia | 0.314 ** | 0.124 [a] (6) |
| Ulmus | 0.242 ** | - |
| Urticaceae | Urtica 0.773 ** , Parietaria 0.609 ** | Urtica 0.642 [a] ** (102), Parietaria 0.445 ** (102) |

\* $p<0.05$, \*\* $p<0.01$, [a] Pearson correlation coefficient

Despite the sensitivity of the fluorescence detectors increased in "middle mode", which expectedly improved representativity
of the Rapid-E+ measurements, some of the clear peaks (e.g. Platanus, Broussonetia) were still not detected (Fig. C2). The
increase of the fluorescence sensitivity also increased fluorescence at shorter wavelengths that dominated in the class "other"
(Fig. E1). This could lead to difficulties in discriminating pollen from other bioaerosols and an additional uncertainty affecting
the discrimination between different pollen classes, in agreement with the confusion matrix of the test dataset (Fig. 2B).

**3.3 Compatibility of different devices and transferability of the classification algorithm**

Rapid-E+ is delivered without a particle classification algorithm and reference pollen datasets, therefore a major effort is
needed to create these monitoring pre-requisites. Repeating it for each device of a network is unfeasible, which puts tight
requirements to compatibility of the measurement signal across devices: an algorithm developed and trained for one device
must be equally (or with minor losses in fidelity) applicable to all devices in the network. At the same time, individual features
of lasers and detectors, as well as variations in the hardware setup resulting in slightly different light paths for different devices,
cause various device-specific features of the signal. As a result, classification performance falls when a model trained on a
reference dataset from one device is tested on a reference dataset from another one (Matavulj et al. 2021). Demonstrated for
Rapid-E, the problem also existed for Rapid-E+ (Fig. 4). The algorithm created on the training dataset collected with the Novi
Sad device failed to identify the same reference pollens collected with both Osijek and FMI devices (F1 score = 0.01 in both
cases).




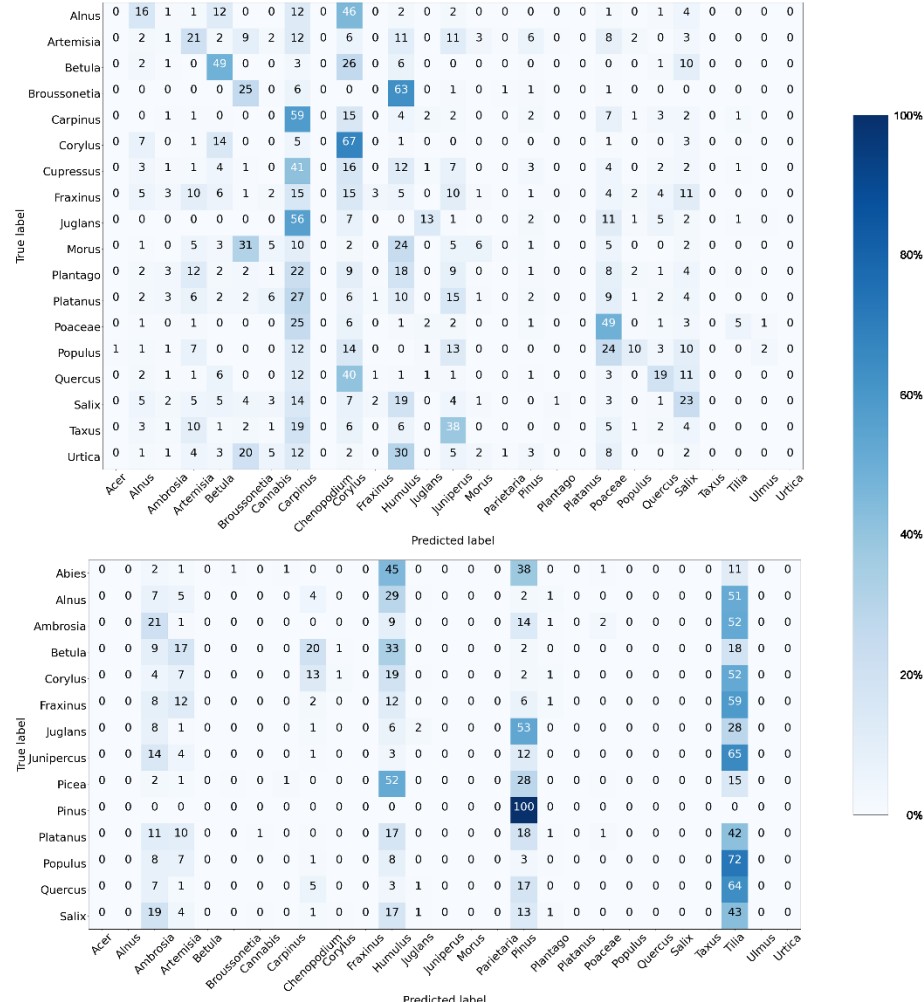

**Figure 4: Confusion matrices with the results depicting performance of classification model trained on the reference dataset collected with the Novi Sad Rapid-E+ device when classifying the same pollen measured on Osijek (upper panel) and FMI (lower panel) devices.**

## 3.4 Strength of the fluorescence signal and difference between devices

Cleaning the reference data based on fluorescence intensity reveals differences in the signal strength between different pollen types, in line with observations from Rapid-E (Smith et al., 2022). This limits detection of pollen with low fluorescence signatures by Rapid-E+. As shown by earlier excitation-emission measurements (Pöhlker et al., 2013), the excitation with the 337 nm laser may lead to a low-intensity response for some pollen types. The most affected pollens are from Pinaceae and Betulaceae families (Table 1).



Analysing the cleaning results of the reference data for the same reference pollen measured with different devices, we noticed a significant difference between the devices for most pollen types. Not all though, there are pollen classes with comparable results across the devices (i.e. Platanus, Salix, Betula), i.e., different timing of the lab work and different methods of exposing the device to pollen cannot explain the differences: each lab worked the same way for all pollen types. Therefore, we conclude

that devices differ in their sensitivity to the scattering and/or fluorescence signals.

When comparing the Betula size measured by Rapid-E+, derived from 447 nm laser scattering image (Fig. 5A and 5B), the distributions are similar for all tested devices (Fig. 5C) but there is a shift between them. Also, the absolute value is smaller than the expected size (10-25 µm) for this pollen grain (Halbritter at al., 2020). This discrepancy could originate from the fact that the linear regression function for calculating the size supplied by the Rapid-E+ manufacturer (Eq. D1) is derived from

measurements of PSLs, which have different refraction characteristics and are more homogenous than pollen. This could also be the reason for negative size reported for some particles, which is an evident artefact, especially since size was positively correlated with intensity of the scattering measurements of Rapid-E (Lieberherr et al., 2021). There is also a big difference between the devices in the average 647 nm laser scatter signals (Fig. 5D).

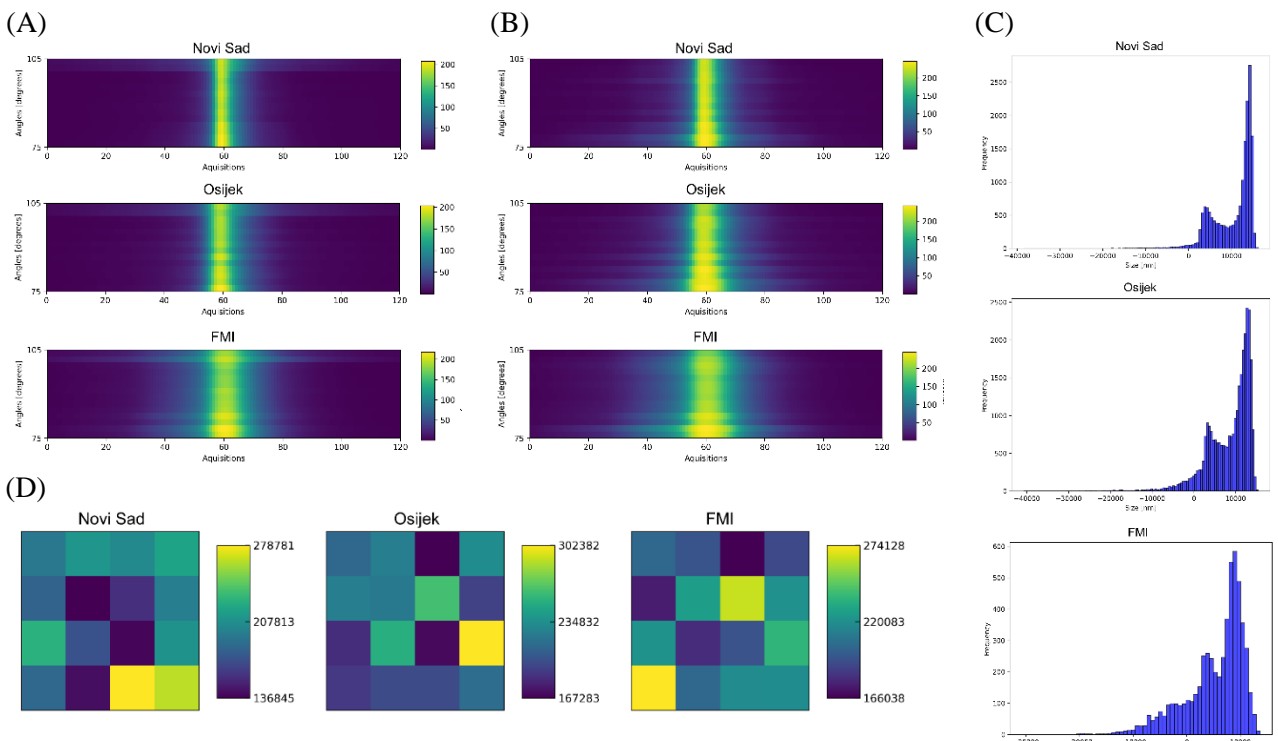

**Figure 5: Comparison of reference Betula pollen measurements in "pollen mode" on Novi Sad, Osijek and FMI Rapid-E+ devices after preprocessing: (A) average 447 nm laser perpendicular polarisation scatter, (B) average 447 nm laser parallel polarisation scatter, (C) histogram of size distribution (D) average 647 nm laser scatter.**





With respect to fluorescence, the difference between devices in the spectrum measurements are hardly noticeable (Fig. 6A).
However, signals of the fluorescence lifetime notably differ (Fig. 6B). The noise seems to dominate in the Betula pollen
average fluorescence lifetime signals from both Osijek and FMI devices (Fig. 6B). Similar differences in the fluorescence
lifetime measurements by different devices are seen also for other directly comparable pollen classes (Figure F1).
These observations explain the poor transferability of the recognition algorithm.

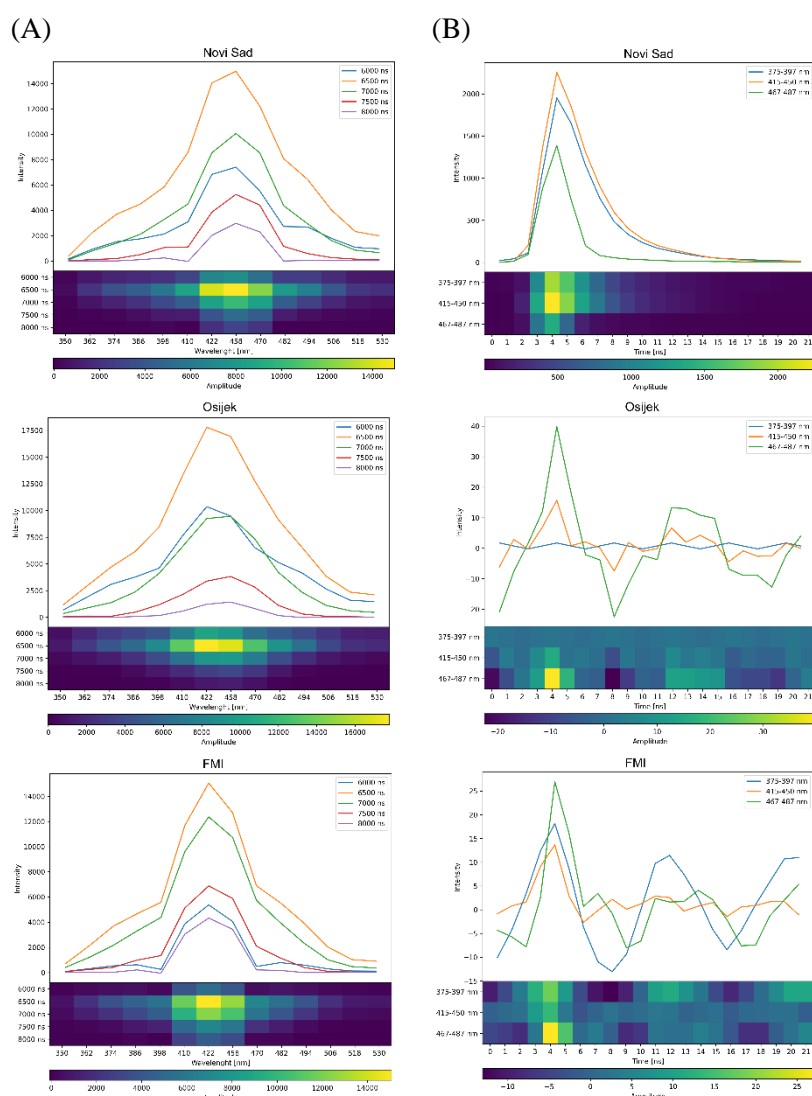

**Figure 6: Comparison of reference Betula pollen fluorescence measurements in "pollen mode" on Novi Sad, Osijek and FMI Rapid-E+ devices after preprocessing: (A) average spectrum, (B) average lifetime. Both regular and image-like smoothed and normalised format used as input for neural network formats used by neural network are presented.**



## 4 Conclusions

The upgrade of Plair Rapid-E to Rapid-E+ brought some improvements in performance regarding identification of pollen and provided some new capabilities. The most-useful new feature is recording the particles in different size bins even when the fluorescence-inducing laser was not activated. Accuracy of the size determination, however, may depend on pollen type, especially for particles that significantly differ from PSLs used for establishing the relationships between the scattering measurements and particle size. The new device worked reliably in continuous measurements and, according to the manufacturer, lifetime of the 337 nm fluorescence inducing laser has been doubled. Ability to detect particles with different sensitivity of fluorescence measurements potentially enables measurements of fungal spores.

The most significant problem we faced with the fluorescence measurements. Uncertainly of the single-particle fluorescence measurements were large, which limited the accuracy of the particle recognition, both in the lab and in the field campaign. At the same time, there is a large discrepancy between the signals measured by different devices. Both aspects make the device unsuitable for large operational monitoring networks: the Rapid-E+ comes without a classification algorithm and training datasets, whose creation is a highly demanding process. Each of the devices analysed in the current study required a full-scale independent training of the algorithm prior to application.

Additional efforts from the manufacturer are needed to increase the signal to noise ratio of the fluorescence measurement, for a wide spectrum of bioaerosols of interest. This is particularly emphasized for regions where numerous pollen and fungal spore classes are simultaneously present in the atmosphere. A much closer collaboration between the manufacturer and its clients is needed to bring Rapid-E+ to the level required for monitoring in operational aerobiological networks.





# Appences A-F

# Appendix A

Table A1: Side by side key specification of PLAIR SL Rapid-E and Rapid-E+ as extracted from user manuals

| Parameter | Rapid-E | Rapid-E+ | |
|---|---|---|---|
| Particle size range, micrometers (μm) | 1-100 | 0.3-100 | |
| Maximum counts, particles per liter | 1600 (fully characterized) | 1000000 (scattering only) 4800 (fully characterized) | |
| Sample air flow, liters per minute (LPM) | 2.8 | 5 | |
| Power supply:<br>Volts AC<br>Volts DC | 90-240<br>18-30 | 90-240 | |
| Power consumption, watts | 200 | 200 | |
| Size (H x W x D), centimeters | 40 x 34 x 73 | 40 x 34 x 55 | |
| Weight, kilograms | 20 | 25 | |
| | | | |
| Scattering laser wavelength, nanometers (nm) | 450 | 447 ± 5 | |
| Scattering image | 24 detectors (each different angle 45-135 degrees) | 2 (perpendicular and parallel polarizations) x 14 detectors (each different angle 75-100 degrees) | |
| Red laser wavelength, nanometers (nm) | - | 637 ± 5 | |
| "infra-red" image | - | 4 x 4 detectors | |
| UV laser wavelength, nanometers (nm) | 337 | 337 ± 5 | |
| Fluorescence spectral range, nm | 350-800 (14 nm per pixel) 32 detectors, 8 records in time (500 ns difference) | 390-570 ± 5 (12 nm per pixel) * 16 detectors, 32 records in time (500 ns difference) | |
| Fluorescence spectral range of lifetime module (nm) | 350-400<br>420-460<br>511-572<br>672-800 | one photodetector per spectral range | 375-397 ± 5<br>415-450 ± 5<br>467-487 ± 5 | one photodetector per spectral range |
| Fluorescence decay resolution, nanoseconds (ns) | 2 (for each spectral range) | 1 (for each spectral range) but two consecutive records are the same value | |

* There is discrepancy in the ranges given in different parts of the Rapid-E+ Operation and Service Manual version 6.2 In the specification on page 9 and in table on page 14 it writes 390-570 nm, in figure on page 14 it is 350-about 560 nm (so resolution is about 14 nm), in figures on pages 21, 22, 24, 28, 29, 30, it is from 350 nm and resolution is larger than 12 nm while in figure on page 31 it is from 350-700 nm (so resolution is 23.34 nm)






Table A2: Pollen classes in tests and results of cleaning the dataset for each device involved in this study. (If more than one species used as pollen source, taxa from which reference data is collected on different devices is marked using bold font.)

| Class label | Pollen source | total number of measured particles (% remaining after cleaning) | | | |
|---|---|---|---|---|---|
| | | Novi Sad pollen mode | Novi Sad middle mode | Osijek pollen mode | FMI pollen mode |
| Abies | *Abies concolor* (Gordon) Lindley ex Hildebrand | - | - | - | 8501 (18%) |
| Acer | *Acer negundo* L. | 7758 (63%) | 3807 (61%) | - | - |
| Alnus | *Alnus glutinosa* L. (Gaertn.) | 14346 (23%) | 12177 (38%) | 11099 (40%) | 53073 (49%) |
| Ambrosia | *Ambrosia artemisiifolia* L. | 23558 (20%) | 17941 (37%) | - | 10973 (45%) |
| Artemisia | *Artemisia absintium* L., **Artemisia vulgaris L.** | 18368 (18%) | 21216 (31%) | 626 (37%) | - |
| Betula | *Betula pendula* Roth | 18089 (21%) | 30240 (14%) | 30531 (29%) | 5667 (25%) |
| Broussonetia | *Broussonetia papyrifera* (L.) Vent. | 7462 (32%) | 6172 (46%) | 16409 (65%) | - |
| Cannabis | *Cannabis sativa* L. | 13049 (33%) | 11013 (31%) | - | - |
| Carpinus | *Carpinus betulus* L. | 11666 (4%) | 13613 (8%) | 9585 (16%) | - |
| Chenopodium | *Chenopudium album* L. | 3441 (12%) | 10522 (16%) | - | - |
| Corylus | **Corylus avellana L.,** *Corylus colurna* L. | 12660 (20%) | 19137 (40%) | 16156 (34%) | 41367 (46%) |
| Cupressus | *Cupressus sempervirens* L. | - | - | 9605 (24%) | - |
| Fraxinus | *Fraxinus angustifolia* Vahl, **Fraxinus pennsylvanica Marshall** | 55921 (19%) | 22673 (65%) | 4334 (30%) | 13782 (56%) |
| Humulus | *Humulus lupulus* L. | 10475 (18%) | 10103 (35%) | - | - |
| Juglans | **Juglans regia L.,** *Juglans nigra* L. | 27507 (20%) | 18497 (45%) | 11512 (21%) | 12459 (38%) |
| Juniperus | *Juniperus virginiana* L. | 9869 (15%) | 65516 (6%) | - | 15600 (58%) |
| Morus | *Morus alba* L. | 30327 (43%) | 6748 (52%) | 7359 (59%) | - |
| Parietaria | *Parietaria officinalis* L. | 10022 (32%) | 11712 (24%) | - | - |
| Picea | *Picea omorica* (Pančić) Purk. | - | - | - | 12963 (18%) |
| Pinus | **Pinus silvestris L.,** *Pinus nigra* Arnold | 37498 (4%) | 85241 (6%) | - | 5175 (43%) |
| Plantago | *Plantago lanceolata* L. | 16882 (38%) | 14829 (63%) | 2627 (47%) | - |
| Platanus | *Platanus orientalis* L. | 7675 (61%) | 12505 (91%) | 7437 (60%) | 15905 (56%) |
| Poaceae | **Dactylis glomerata L.,** *Poa trivialis* L., **Dasypyrum villosum (L.) Borbás** | 19536 (52%) | 40624 (45%) | 13624 (67%) | - |
| Populus | *Populus alba* L., **Populus canadensis Moench.,** **Populus nigra L.,** *Populus nigra var. pyramidalis* Spach. | 20844 (28%) | 23880 (58%) | 9705 (44%) | 54803 (76%) |
| Quercus | **Quercus robur L.,** *Quercus robur var. pyramidalis* C.C.Gmel. | 36114 (16%) | 28132 (44%) | 11738 (28%) | 27351 (41%) |
| Salix | **Salix alba L.,** **Salix caprea** L. | 9740 (32%) | 8183 (65%) | 3163 (38%) | 5061 (34%) |
| Taxus | *Taxus baccata* L. | 16801 (25%) | 23301 (9%) | 9320 (50%) | - |
| Tilia | *Tilia tomentosa Moench.* | 11836 (16%) | 25917 (43%) | - | - |
| Ulmus | *Ulmus sp.* | 4211 (53%) | 8549 (23%) | - | - |
| Urtica | *Urtica dioica* L. | 4537 (64%) | 14281 (65%) | 5437 (43%) | - |





# Appendix B

(A)

(B)

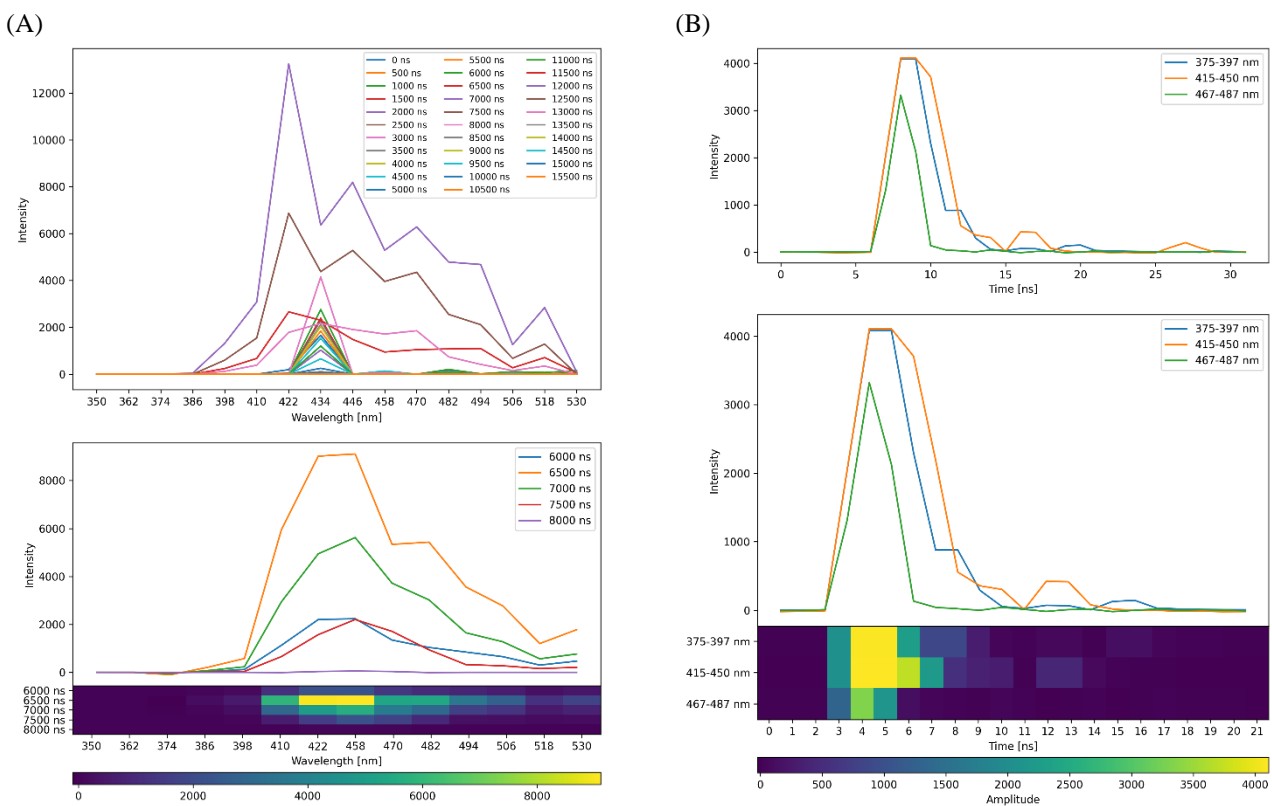

Figure B1: Preprocessing of Rapid-E+ single Betula pollen fluorescence measurements: (A) spectrum and (B) lifetime. Upper panel raw signal, middle panel selection of suitable measurements from raw signal, lower panel image-like smoothed and normalised format used as input for neural network. (y-axis is „unitless")



(A)

(B)




Figure B2: Average (with standard deviation depicted by area around lines) fluorescence spectrum (left side) and lifetime (right side) measurements after preprocessing for: (A) *Betula pendula,* (B) *Fraxinus pennsylvanica,* (C) *Juglans regia* and (D) *Platanus orientalis* refference pollen measured in „pollen mode" on Novi Sad Rapid-E+ device. (y-axis is „unitless")




# Appendix C

(A)  (B)

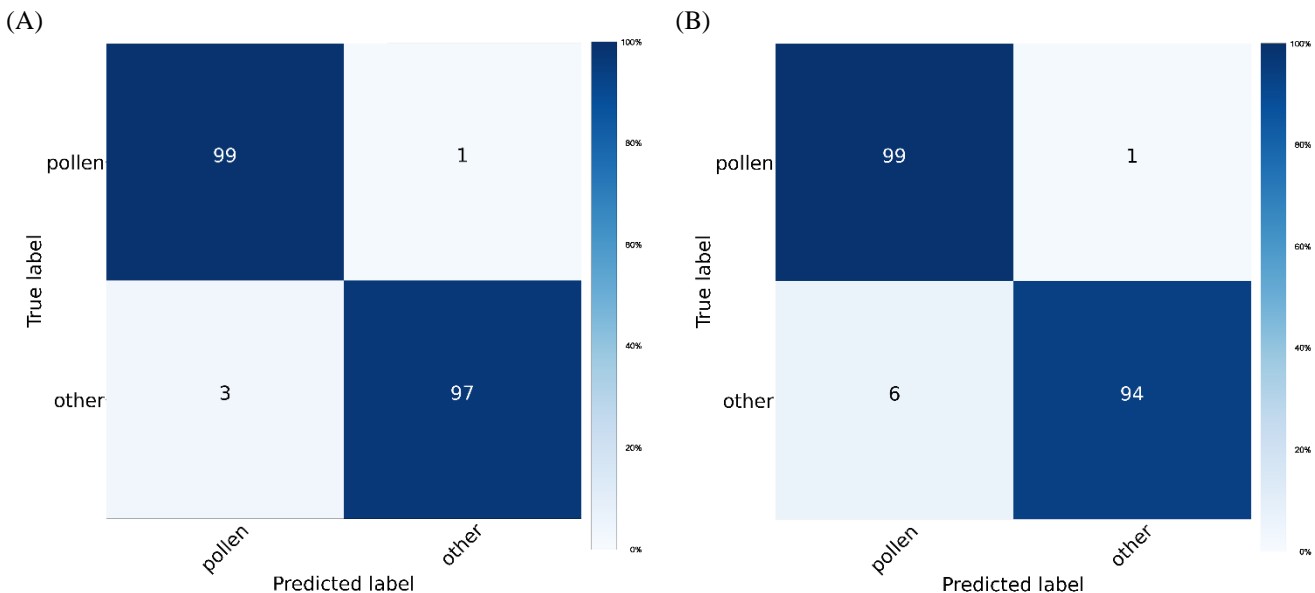

Figure C1: Confusion matrices depicting performance of classification model in discriminating pollen from other bioaerosols on test dataset measured in (A) "pollen mode" and (B) "middle mode".






Figure C2: Daily pollen concentrations measured side-by-side using Rapid-E+ device (orange) and standard EN16868 method (blue) for pollen classes with concentrations exceeding 10 pollen m$^{-3}$ at least 10 days. (Mind the difference in y-axes). Rapid-E+ records affected by collecting refference datasets and interruption in measurements were removed.





















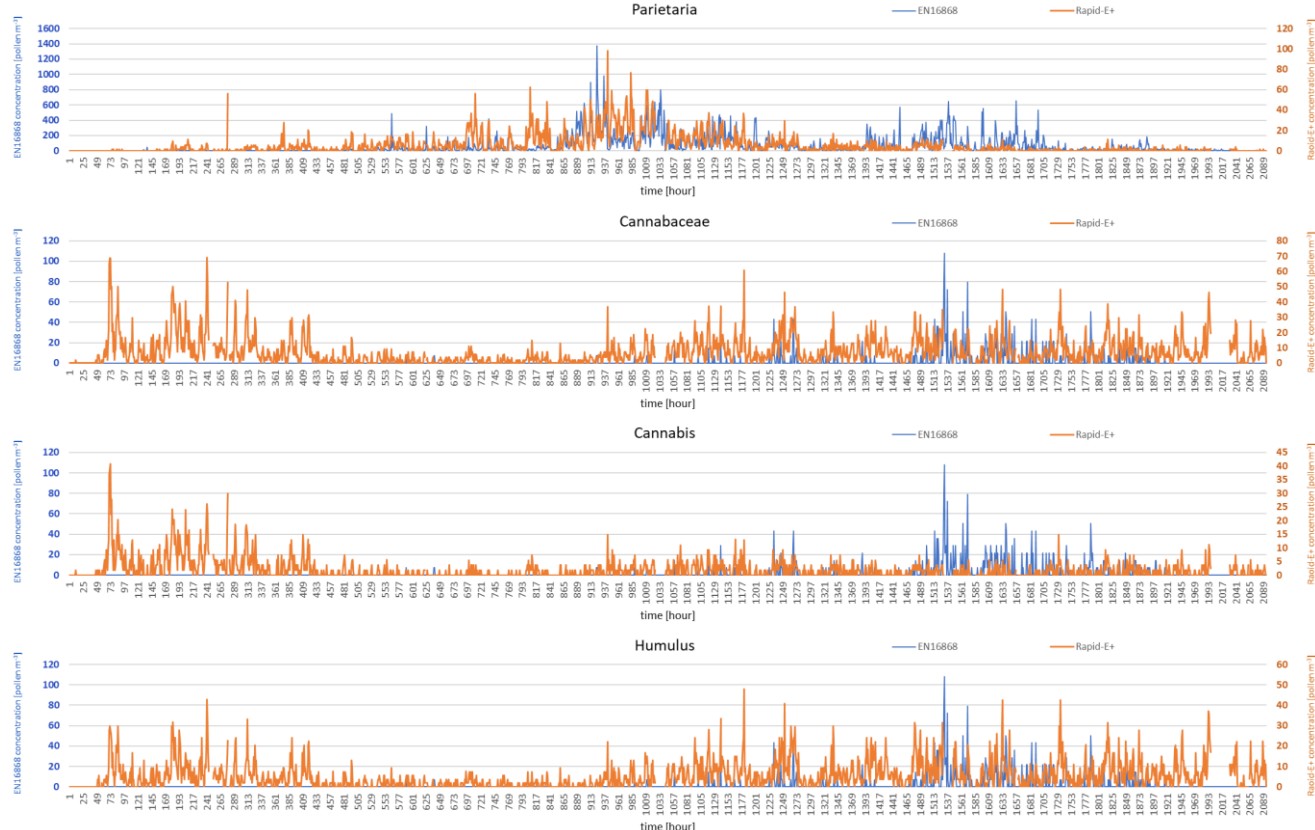

Figure C3: Two-hourly pollen and total fungal spores concentrations measured side-by-side using Rapid-E+ device (orange) in „pollen mode" and standard EN16868 method (blue). (Mind the difference in y-axes.) Rapid-E+ records affected by collecting refference datasets and interruption in measurements were removed.





# Appendix D


*Equation* (1) for calculating particle size in nm using features derived from scattering measurements. The formula is supplied by manufacturer that claimed it is obtained from data collected during device calibration at Swiss Federal Institute of Metrology

SizeEstimate [in nm] = USum * (-6.86543945e-04) + DSum * (-2.25737316e-04) + Umax * 9.32852833e-03 +DMax * (1.12780948e-02) + UDur * (4.78690614e+00) + DDur * (5.73476089e-01) + 422.01107368773455)

      (1)

    USum: sum of all pixels of the perpendicular polarization

    DSum: sum of all pixels of the parallel polarization

    UMax: maximum of the sum of all pixels of the perpendicular polarization

    DMax: maximum of the sum of all pixels of the parallel polarization

    UDur: duration of the perpendicular polarization

    DDur: duration of the parallel polarization



# Appendix E

(A)

(B)

(C)

(D)

(E)

(F)

(G)

(H)

(I)

(J)



(K)

(L)

(M)

(N)

(O)

(P)

(R)

(S)

(T)

(Q)






Figure E1: Average normalized fluorescence spectrum in „pollen mode" (left side) and „middle mode" (right side) measured using Novi Sad Rapid-E+ device for refference pollen: (A) Acer, (B) Alnus, (C) Ambrosia, (D) Artemisia, (E) Betula, (F) Cannabis, (G) Carpinus, (H) Chenopodium, (I) Corylus, (J) Fraxinus, (K) Humulus, (L) Juglans, (M) Morus, (N) Broussonetia, (O) Urtica, (P) Parietaria, (R) Poaceae, (S) Populus, (T) Quercus, (Q) Salix, (W) Taxus, (X) Juniperus, (Y) Tilia, (Z) Pinus, (AA) Ulmus, (AB) Plantago, (AC) Platanus, (AD) „other". (y-axis is „unitless").



# Appendix F

(A)                                (B)                                (C)

Figure F1. Comparison of (A) Fraxinus, (B) Juglans and (C) Platanus pollen average fluorescence lifetime measurements in "pollen mode" after preprocessing, on Novi Sad, Osijek and FMI Rapid-E+ devices. Both regular and normalized image-like formats used by neural network are presented. (y-axis is „unitless")


**Authors contribution**

Branko Sikoparija: Conceptualization, Observational data preparation, processing and evaluation, Formal analysis, Writing - original draft, financial, managerial, and administrative support, Predrag Matavulj: Formal analysis, Isidora Simovic: Observational data preparation, processing and evaluation, Predrag Radisic: Observational data preparation, processing and evaluation, Sanja Brdar: Formal analysis, Vladan Minic: Formal analysis, Danijela Tesendic: Formal analysis, Evgeny Kadantsev: Observational data preparation, processing and evaluation, Julia Palamarchuk: Observational data preparation, processing and evaluation, Mikhail Sofiev: Writing, reviewing and editing, financial, managerial, and administrative support.

**Competing interests**

The authors declare no competing interests.

**Code and data availability**

The training data and the machine learning algorithm are openly available from the authors upon request (note the low transferability of this information).

**Acknowledgements**

The authors acknowledge support of the Horizon project SYLVA (grant no. 101086109) and Ministry Science, Technological Development and Innovations of the Republic of Serbia (Grant agreement no. 200358).

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
