# Peer review of "Classification accuracy and compatibility across devices of a new Rapid-E+ flow cytometer"

_EGUsphere, 2024_

## Referee Comment (RC1)

This manuscript "Classification accuracy and compatibility across devices of a new Rapid-E+ flow cytometer" describes the evaluation of a new instrument, the Rapid-E+, upgraded from a previous model made by Plair SA, and its ability to monitor pollen compared alongside a manual Hirst-type sampler. The necessary training of a classification algorithm to distinguish pollen types is detailed and lab evaluation is followed up by field evaluation, and cross-comparison with instruments at other sites to assess method generalisability. The study is thorough and comprehensive, looking into the detail of the different modalities of data obtained for different pollen types across different instruments.

The manuscript is of rigorous scientific quality and reports findings that are useful in this field to further the advancement of automated pollen monitoring. It is written and presented concisely and generally clearly, with ample supporting information in the Appendices. There are only some minor technical points that I would address before continuing to publication.

Please see below for specific comments by line.

**Abstract**

Line 22: I would use the term 'instrument' instead of 'monitor'.

**Introduction**

Line 29: "Buters et al. 2022"

Line 30: "monitoring instruments"

**Materials and Methods**

Line 49-50: Not sure in this sentence exactly how the Rapid-E+ compares to the Rapid-E. Perhaps alter to "In particular the Rapid-E+ samples at a faster flow rate of 5 l min$^{-1}$ (compared to 2.8 l min$^{-1}$ for the Rapid-E), and records all particles passing through a 447 nm scattering laser into 4 size bins (>0.3 μm, >0.5 μm, >1 μm, >5 μm) unlike the Rapid-E which...?" (does the Rapid-E not have different size bins?)

Line 55-56: "also allows for adjusting the gain of the fluorescence spectrum and lifetime detectors"

Line 72: "Three Rapid-E+ air flow cytometers were involved in this study."

Line 72: "...in Novi Sad, Serbia, ..."

Line 73: "the Novi Sad laboratory" is very nondescript. Details about the organisation that runs the Novi Sad laboratory may be helpful, and the environment?

Line78: "The test period allowed for the exploration of measurement performance of the automatic bioaerosol monitoring instrument in a variety of conditions characteristic of the Pannonian Plain in [where?]. This region contains a large diversity of pollen and fungal spores..." This sentence was quite long so I suggest splitting it into two, e.g. where I have done so.

Line 82: "the period of seasonal allergies" – perhaps a little more description specifically as to what these seasonal allergies are in this place?

Line 83: "when large quantities of ragweed pollen are recorded in the air"

Line 85: "the main features of diurnal variations"

Line 89: "Reference pollen for training was collected locally."

Line 98: "to ensure identity" - could you explain this better?

Line 102: "exposed to pollen using the Swisens Atomizer"

Line 103: "expose pollen to the Novi Sad and Osijek devices.

Line 106: "validating"

Line 109: Could say "colocated" instead of side-by-side.

**Results and discussion**

Line 201: Are these precision, recall and F1 scores averaged across scores for each pollen classification? If so, just mention they are averaged to avoid confusion, if not, I am unsure how the score differs from the discrimination of pollen from "other".

Line 207: By 'the classification algorithm with high accuracy' do you mean the one that achieved F1 score of 0.86 as opposed to 0.84? Or simply that the algorithm managed to distinguish these pollen types with high accuracy, regardless as to which? Perhaps it may be better to write something like one of the following, depending on which you meant to avoid confusion...

"It is interesting to note that the latter classification algorithm (with merged classes) distinguished Urtica and Parietaria from Brousonetia despite these pollen grains being morphologically similar."

Or

"It is interesting to note that the classification algorithm distinguished Urtica and Parietaria from Brousonetia with high accuracy, despite these pollen grains being morphologically similar."

Fig. 2: The numbers and names are a bit small and blurry, would be good to make the characters a little larger if possible.

Line 226: what are the exact dates referred to here?

Line 235: Best to define PSLs in brackets for good measure as it is mentioned for the first time in this manuscript.

Line 241: At a glance, this sentence was a little confusing, I would correct it to something like: "Automatic detections of total pollen, as well as Juglans, Morus and Ambrosia, have a statistically significant positive correlation with…"

Line 243: "for most pollen classes" or "for most of the pollen classes"

Line 245: Perhaps rephrase as, for example, "Pollen grains that occur simultaneously in the air had a clear tendency to be confused amongst each other, which was expected…"

Line 261: "As demonstrated for the Rapid-E, this problem also exists for the Rapid-E+."

Line 278: I would probably start a new sentence and replace the second i.e. before 'different timing…' with something else. This sentence is a bit confusing and long. Is it saying that since some pollen classes were comparable across devices, the differences observed across others shouldn't be due to doing lab work at different times and different methods of pollen exposure to the instrument? Or are you saying each lab followed the same procedures so it shouldn't be an issue?

Fig. 5 writing font too small and am unsure what I am looking at in 5D, can labels be added to the x, y and colour axes?

Fig. 6 again writing font too small.

---

## Author Comment (AC1)

**Classification accuracy and compatibility across devices of a new Rapid-E+ flow cytometer**

Branko Sikoparija[1], Predrag Matavulj[2], Isidora Simovic[1], Predrag Radisic[1], Sanja Brdar[1], Vladan Minic[1], Danijela Tesendic[3], Evgeny Kadantsev[4], Julia Palamarchuk[4] and Mikhail Sofiev[4]

[1]BioSense Institute Research Institute for Information Technologies in Biosystems, University of Novi Sad, Novi Sad, 2100, Serbia

[2]Institute for Data Science, University of Applied Sciences North Western Switzerland, Windish, 5210, Switzerland

[3]Department of mathematics and informatics, Faculty of Sciences, University of Novi Sad, Novi Sad, 21000, Serbia

[4]Finnish Meteorological Institute, Helsinki, Erik Palmenin Aukio 1, FI-00560, Finland

*Correspondence to*: Mikhail Sofiev (Mikhail.Sofiev@fmi.fi)

Matt Smith #1 (Citation: https://doi.org/10.5194/egusphere-2024-187-CC1)

The authors present a very interesting and robust study examining the classification accuracy and compatibility across devices of a new Rapid-E+ flow cytometer for examining airborne pollen. The paper is generally well written, although it could do with thorough editing with specific focus on the use of articles. I have listed some minor comments below that I hope will help.

**Reply:** The authors would like to thank Matt Smith for interest in the study and for his helpful comments, which we have used to improve our manuscript. Below we answer the questions and indicate the changes we have made to the revised manuscript.

My one comment about the methods relates to the use of the Hirst type trap (Lines 161 to 165). When calibrating such sensitive instruments as the Rapid-E and Rapid-E+, it is important to remove as much uncertainty as possible. The authors might therefore consider counting whole slides from the Hirst type trap to reduce error. Obviously, this is not always feasible when examining whole seasons, but even examining a small subset of slides in this way might provide some interesting insights. Although I note that correlations were only conducted for or days when average pollen concentrations measured by the manual method exceeded 10 pollen $m^{-3}$ in order to reduce uncertainty.

**Reply:** The reviewer is correct that the standard method (EN16868) has large uncertainty that originates from different critical points (i.e. flow measurements, pollen identification, subsampling during analysing collected samples). Recent study by Mimic and Sikoparija (2021) confirmed that analysing 100% of samples coming from Hirst type traps is expected to improve comparison of time series obtained from different devices especially for low concentrations. However, as the reviewer correctly pointed out, analysing an entire sample under microscope for the entire season is nor realistic, the effect is quite small, and all measurement critical points exist in an automatic approach as well (Tummon et al. 2022), but still are not precisely quantified. This is why we followed the recommendations of the EN16868 norm: to limit the influence of measurement uncertainty when comparing results from different methods. We focused on daily values and only considered cases where a sufficient amount of pollen was detected. To clearly address this aspect, we have added the following sentence to section 3.3:

"Limited improvement in correlations could be expected if the measurement uncertainty of the standard Hirst volumetric method (EN16868), inherited from the subsampling during analysing collected samples, is eliminated by counting 100% of microscopic slides (Mimic and Sikoparija 2021). However, such analysis for the entire season is extremely difficult and even if done so, the effect is presumed to be small."

Minor comments

Line 47 - "which is a new model stemming from the PA-300 (Crouzy et al., 2016) and Rapid-E (Sauliene et al., 2019)".
**Reply:** Corrected as suggested.

Line 49 – "In particular, Rapid-E+ samples at a flow rate of 5 l min-1"
**Reply:** Corrected as suggested.

Line 53 – "Like its predecessor"
**Reply:** Corrected as suggested.

Line 73 – "was trained in the Novi Sad laboratory"
**Reply:** Corrected as suggested.

Line 74 – "owned by the City of Osijek in Croatia"
**Reply:** Corrected as suggested.

Line 74 – "and the Finnish Meteorological Institute"
**Reply:** Corrected as suggested.

Lines 79/80 – "for the Pannonian Plain" Lines 85/86 – "or capturing the main features"
**Reply:** Corrected as suggested.

Line 91 – "Scientific names should be italics" (review throughout including figures and tables).
**Reply:** The scientific names of the plant species from which pollen was used in the model training were written in italics (Table A2). For classes of pollen identified from aerobiological samples (automatic and manual) we did not use the taxonomic nomenclature because the pollen classes do not fully represent taxonomic categories. For example, in real time detections class Artemisia is trained on pollen from *Artemisia absintium* L*., Artemisia vulgaris* L., thus it cannot be fully representative for genus *Artemisia*. Similarly, in manual analysis the class Artemisia recorded in the given day could consist of pollen coming either from one or several species thus never being representative for the entire genus *Artemisia*. To address this, we have added the following info in the Table A2:
"* does not fully represent taxonomic rank (i.e. pollen in reference data coming only from one or several species of the respective taxonomic category) thus not written in italics"

Lines 98/99 – "To ensure identification"
**Reply:** Corrected as suggested

Line 102 - by using a Swisens Atomizer
**Reply:** Corrected as suggested

Line 193 – "It is interesting to note that after the start of rainfall the coarse particles"
**Reply:** Corrected as suggested.

Line 196 – The following lacks clarity and should be rewritten "However, quite low flow rate"
**Reply:** The sentence is rewritten and now reads:
"However, following the equations given in Tummon et al. (2022), the flow rate of the Rapid-E+ (5 l min$^{-1}$) is not sufficient to measure all relevant concentrations at sub hour temporal resolution with reasonably low uncertainty."

Line 208 – "despite these pollen grains being morphologically similar" (note that the plural of pollen is pollen)
**Reply:** Corrected as suggested.

Line 245 – "There was a clear tendency towards confusion of different pollen occurring"

**Reply:** Corrected as suggested.

Table 1 - It would be interesting to see the correlation coefficients for Taxaceae/Cupressaceae combined and for the Urticaceae
family, as many pollen monitoring networks do not separate these into different genera due to the difficulty in identification.
**Reply:** We have calculated correlations for Taxaceae/Cupressaceae (sum of Taxus and Juniperus in Rapid-E+ data), Urticaceae
(sum of Urtica and Parietaria in Rapid-E+ data) and Cannabaceae (sum of Cannabis and Humulus in Rapid-E+ data), and
added coefficients into Table 1. Also, we added the following sentence to the results section:
"Merging Rapid-E+ measurements for classes that are difficult to identify by manual method (i.e. Taxus and Juniperus, Urtica
and Parietaria, Cannabis and Humulus) did not improve the correlations (Table 1)."

Line 256 – "Repeating it for each device in a network is unfeasible"
**Reply:** Corrected as suggested.

Lines 261/262 – The following text lacks clarity and needs reworking, perhaps linked to another sentence "Demonstrated for
Rapid-E, the problem also existed for Rapid-E+ (Fig. 4)".
**Reply:** The text is now rewritten and reads:
"As a result, classification performance falls when a model trained on a reference dataset from one device is tested on a
reference dataset from another one, which was demonstrated for Rapid-E (Matavulj et al. 2021). The same problem exists in
Rapid-E+ (Fig. 4)."

Line 263 - pollen not pollens
**Reply:** Corrected as suggested.

Line 274 - pollen not pollens
**Reply:** Corrected as suggested.

Line 277 – "Although this was not seen for all pollen types, there are pollen classes with comparable"
**Reply:** Corrected as suggested.

Line 317 – "datasets, the creation of which is a highly demanding process".
**Reply:** Corrected as suggested.

**References**
Mimic, G., Sikoparija, B.: Analysis of airborne pollen time series originating from Hirst-type volumetric samplers—
comparison between mobile sampling head oriented toward wind direction and fixed sampling head with two-layered
inlet, Aerobiologia 37:321-331, https://doi.org/10.1007/s10453-021-09695-7, 2021.
Tummon, F., Bruffaerts, N., Celenk, S., Choël, M., Clot, B., Crouzy, B., Galán, C., Gilge, S., Hajkova, L., Mokin, V.,
O'Connor, D., Rodinkova, V., Sauliene, I., Sikoparija, B., Sofiev, M., Sozinova, O., Tesendic, D. and Vasilatou, K.:
Towards standardisation of automatic pollen and fungal spore monitoring: best practises and guidelines, Aerobiologia.
https://doi.org/10.1007/s10453-022-09755-6, 2022.

---

## Author Comment (AC2)

**Classification accuracy and compatibility across devices of a new Rapid-E+ flow cytometer**

Branko Sikoparija[1], Predrag Matavulj[2], Isidora Simovic[1], Predrag Radisic[1], Sanja Brdar[1], Vladan Minic[1], Danijela Tesendic[3], Evgeny Kadantsev[4], Julia Palamarchuk[4] and Mikhail Sofiev[4]

[1]BioSense Institute Research Institute for Information Technologies in Biosystems, University of Novi Sad, Novi Sad, 2100, Serbia
[2]Institute for Data Science, University of Applied Sciences North Western Switzerland, Windish, 5210, Switzerland
[3]Department of mathematics and informatics, Faculty of Sciences, University of Novi Sad, Novi Sad, 21000, Serbia
[4]Finnish Meteorological Institute, Helsinki, Erik Palmenin Aukio 1, FI-00560, Finland

*Correspondence to*: Mikhail Sofiev (Mikhail.Sofiev@fmi.fi)

Anonymous Referee #2 (Citation: https://doi.org/10.5194/egusphere-2024-187-RC2)

A new device from Plair SA company Rapid-E+ is investigated in current study. A two-step classification was applied. At the first step of classification pollen are separated from non-pollen particles. At the second step pollen are classified into 27 pollen classes. It as established, that as with previous device rapid-E remains a large discrepancy between the signals measured by different devices. Therefore individual models need to be trained for every device. In overall the paper is well prepared. Some minors points must be corrected before final publication.

**Reply:** The authors would like to thank Referee #2 for reviewing the manuscript and positive opinions. We are grateful for helpful comments, which we have used to improve our manuscript. Below we answer the questions and indicate the changes we have made to the revised manuscript.

The paragraph about the used model (135-150) should be extended. ResNet-18 has 4 2-layer blocks. What does mean 4-block-layer or 3-block-layer? In context of ResNet style models, a block is a container of layers. It means that a block is a larger unit than a layer. It seems that not all neural networks have 18 layers, because their architectures are different. That to present the architectures to readers, a good point would be to prepare a architecture table as Table 3 in the paper (https://arxiv.org/pdf/1803.06131). It would also be useful to show the size of the inputs arrays received by each mode sub-network.

**Reply:** The paragraph has been extended as requested, and now reads:

"The ResNet architecture with shortcut connections was chosen for its proven superior performance in classifying pollen using Rapid-E measurements (Matavulj et al., 2023; Daunys et al., 2022). Given the variability of input data, we adapted the ResNet model inspired by the 18-layer version. Specifically, we implemented a 4-block layer for the fluorescence spectrum and lifetime, a 3-block layer for the 447 nm laser scattering images, and a 1-block layer for the 637 nm laser scattering image. Details of these configurations are provided in Table B1. These architectures were selected because they demonstrated the best performance for the respective data types in the previous device version (Matavulj et al., 2023). The block-layers contained three convolutional layers, where we captured a residual following the initial convolution. Subsequently, at the closure of each block layer, we established a residual connection to the layer's output. Following the completion of all block layers, an additional convolutional layer was integrated. This was followed by a global average pooling, which averaged over the spatial dimensions of the images. The network initially learned from each type of input separately. After this initial training, we transferred the learned features from these individual inputs (specifically, the parts of the network responsible for feature extraction, known as convolutional blocks) to a new network. This new network processed all different inputs together by equalizing the features from each input using a fully connected (FC) layer, which were then merged. Finally, the network was trained only to classify this combined data using another FC layer with a SoftMax function. During this phase, the weights of the feature extractors (the convolutional blocks) were kept unchanged. This means that while the network was learning to classify the merged data, the initial parts that extract features from each input type did not undergo any further changes."

Table B1: Feature extractors for each data type. The convolutional layers are represented as N x M, F, where N X M represents the filter size for the 2D convolution, while F represents the number of feature maps.

| Input type: | Scattered light images | Fluorescence spectrum | Fluorescence lifetime | Infrared image |
| --- | --- | --- | --- | --- |
| Input dimension: | 120x14 | 5x14 | 3x22 | 4x4 |
| conv1 | 7 x 7, 70 | 1 x 7, 70 | 1 x 7, 70 | 3 x 3, 70 |
| block1 | 3 x 3, 70
x 3, 70
x 3, 70 | 1 x 3, 70
x 3, 70
x 3, 70 | 1 x 3, 70
x 3, 70
x 3, 70 | 3 x 3, 70
x 3, 70
x 3, 70 |
| block2 | 5 x 5, 140
x 5, 140
x 3, 140 | 1 x 7, 140
x 5, 140
x 3, 140 | 1 x 5, 140
x 5, 140
x 3, 140 | |
| block3 | 7 x 1, 200
x 5, 200
x 3, 200 | 1 x 5, 200
x 5, 200
x 3, 200 | 1 x 3, 200
x 5, 200
x 3, 200 | |
| block4 | | 1 x 3, 300
x 5, 300
x 3, 300 | 1 x 3, 300
x 5, 300
x 3, 300 | |
| final_conv | 3 x 3, 200 | 3 x 3, 300 | 3 x 3, 300 | 4 x 4, 70 |

The scattering images of Rapid-E were of variable length. What is case in Rapid-E+? If they are of variable size, how the issue was solved?
**Reply:** The scattering image in Rapid-E+ has a fixed length of 120 acquisitions across 14 scattering angles. We have now noted that in chapter 2.1 "The 447 nm laser scattering is measured now in two polarization planes at a narrower angle window and fixed duration limited to 120 acquisitions."

It would seem that in the graphs shown in Figure B2 of Appendix B, the intensity should be positive. However, a large part of the shadow, which is bounded by the curvatures calculated adding and subtracting standard deviation to/from the mean, is in the negative range. The standard deviation is appropriate to characterize the dispersion when the values follow a normal distribution. In this case, the distribution does not appear to be normal and, moreover, asymmetric. In this case, it is preferable to represent in the center by solid line a median curve and to delimit the shaded area by curves corresponding to quantiles symmetrical with respect to the median.
**Reply:** Figure B2 of Appendix B has been changed accordingly, where a solid line now represents a median and the shaded area represents the interquartile range (25th - 75th percentiles).

(A)

[Figure]

(B)

(C)

[Figure]

(D)

Figure B2: Median (with the interquartile range 25th - 75th percentiles depicted by area around lines) fluorescence spectrum (left side) and lifetime (right side) measurements after preprocessing for: (A) *Betula pendula*, (B) *Fraxinus pennsylvanica*, (C) *Juglans regia* and (D) *Platanus orientalis* reference pollen measured in "pollen mode" on Novi Sad Rapid-E+ device. (y-axis is "unitless")

**References**

Matavulj, P., Panić, M., Šikoparija, B., Tešendić, D., Radovanović, M., and Brdar, S.: Advanced CNN Architectures for Pollen Classification: Design and Comprehensive Evaluation, Applied Artificial Intelligence, 35, 1, e2157593, https://doi.org/10.1080/08839514.2022.2157593, 2023.

Daunys, G., Šukienė, L., Vaitkevičius, L., Valiulis, G., Sofiev, M., and Šaulienė, I.: Comparison of computer vision models in
application to pollen classification using light scattering. Aerobiologia, https://doi.org/10.1007/s10453-022-09769-0,
2022.

---

## Author Comment (AC3)

**Classification accuracy and compatibility across devices of a new Rapid-E+ flow cytometer**

Branko Sikoparija[1], Predrag Matavulj[2], Isidora Simovic[1], Predrag Radisic[1], Sanja Brdar[1], Vladan Minic[1], Danijela Tesendic[3], Evgeny Kadantsev[4], Julia Palamarchuk[4] and Mikhail Sofiev[4]

[1]BioSense Institute Research Institute for Information Technologies in Biosystems, University of Novi Sad, Novi Sad, 2100, Serbia
[2]Institute for Data Science, University of Applied Sciences North Western Switzerland, Windish, 5210, Switzerland
[3]Department of mathematics and informatics, Faculty of Sciences, University of Novi Sad, Novi Sad, 21000, Serbia
[4]Finnish Meteorological Institute, Helsinki, Erik Palmenin Aukio 1, FI-00560, Finland

*Correspondence to*: Mikhail Sofiev (Mikhail.Sofiev@fmi.fi)

Anonymous Referee #1 (Citation: https://doi.org/10.5194/egusphere-2024-187-RC1)

This manuscript "Classification accuracy and compatibility across devices of a new Rapid-E+ flow cytometer" describes the evaluation of a new instrument, the Rapid-E+, upgraded from a previous model made by Plair SA, and its ability to monitor pollen compared alongside a manual Hirst-type sampler. The necessary training of a classification algorithm to distinguish pollen types is detailed and lab evaluation is followed up by field evaluation, and cross-comparison with instruments at other sites to assess method generalisability. The study is thorough and comprehensive, looking into the detail of the different modalities of data obtained for different pollen types across different instruments. The manuscript is of rigorous scientific quality and reports findings that are useful in this field to further the advancement of automated pollen monitoring. It is written and presented concisely and generally clearly, with ample supporting information in the Appendices. There are only some minor technical points that I would address before continuing to publication.

**Reply:** The authors would like to thank Referee #1 for constructive and positive suggestions on how to improve the manuscript further. Below we answer the questions and indicate the changes we have made to the revised manuscript.

Abstract
Line 22: I would use the term 'instrument' instead of 'monitor'.
**Reply:** Corrected as suggested throughout text.

Introduction
Line 29: "Buters et al.  2022"
**Reply:** Corrected as suggested.

Line 30: "monitoring instruments"
**Reply:** Corrected as suggested.

Materials and Methods
Line 49-50: Not sure in this sentence exactly how the Rapid-E+ compares to the Rapid-E. Perhaps alter to "In particular the Rapid-E+ samples at a faster flow rate of 5 l min-1 (compared to 2.8 l min-1 for the Rapid-E), and records all particles passing through a 447 nm scattering laser into 4 size bins (>0.3 µm, >0.5 µm, >1 µm, >5 µm) unlike the Rapid-E which…?" (does the Rapid-E not have different size bins?)

**Reply:** The statement is expanded to compare differences and now reads:

"In particular, Rapid-E+ samples at a faster flow rate of 5 l min$^{-1}$ (compared to 2.8 l min$^{-1}$ for the Rapid-E). Also, regardless the operation mode, Rapid-E+ records concentration of all particles passing through a 447 nm scattering laser (classified into 4 size bins: >0.3 µm, >0.5 µm, >1 µm, >5 µm), while Rapid-E only records concentration of particles above operation mode determined size limit."

Line 55-56: "also allows for adjusting the gain of the fluorescence spectrum and lifetime detectors"
**Reply:** Corrected as suggested.

Line 72: "Three Rapid-E+ air flow cytometers were involved in this study."
**Reply:** Corrected as suggested.

Line 72: "…in Novi Sad, Serbia, …"
**Reply:** Corrected as suggested.

Line 73: "the Novi Sad laboratory" is very nondescript. Details about the organisation that runs the Novi Sad laboratory may be helpful, and the environment?
**Reply:** As suggested, we have specified that device worked indoors during creation of the training dataset and then set to work outside.

Line78: "The test period allowed for the exploration of measurement performance of the automatic bioaerosol monitoring instrument in a variety of conditions characteristic of the Pannonian Plain in [where?]. This region contains a large diversity of pollen and fungal spores…" This sentence was quite long so I suggest splitting it into two, e.g. where I have done so.
**Reply:** Corrected as suggested

Line 82: "the period of seasonal allergies" – perhaps a little more description specifically as to what these seasonal allergies are in this place?
**Reply:** The sentence is extended and now reads:

"In the study region, the period of seasonal pollen allergies (i.e. tree pollen season from January to April and grass pollen season from April to September) is extended by the weed pollen season from July to the end of October when large quantity of ragweed pollen is recorded in the air (Sikoparija et al., 2018)"

Line 83: "when large quantities of ragweed pollen are recorded in the air"
**Reply:** Corrected as suggested.

Line 85: "the main features of diurnal variations"
**Reply:** Corrected as suggested.

Line 89: "Reference pollen for training was collected locally."
**Reply:** Corrected as suggested

Line 98: "to ensure identity" - could you explain this better?
**Reply:** This part was indeed confusing, so we removed it from the text.

Line 102: "exposed to pollen using the Swisens Atomizer"
**Reply:** Corrected as suggested.

Line 103: "expose pollen to the Novi Sad and Osijek devices.
**Reply:** Corrected as suggested.

Line 106: "validating"
**Reply:** Corrected as suggested.

Line 109: Could say "colocated" instead of side-by-side.
**Reply:** Corrected as suggested.

Results and discussion
Line 201: Are these precision, recall and F1 scores averaged across scores for each pollen classification? If so, just mention
they are averaged to avoid confusion, if not, I am unsure how the score differs from the discrimination of pollen from "other".
**Reply:** Yes, that is correct. The F1 scores were calculated for each class and then averaged. It is now indicated in the text, as
suggested.

Line 207: By 'the classification algorithm with high accuracy' do you mean the one that achieved F1 score of 0.86 as opposed
to 0.84? Or simply that the algorithm managed to distinguish these pollen types with high accuracy, regardless as to which?
Perhaps it may be better to write something like one of the following, depending on which you meant to avoid confusion…
"It is interesting to note that the latter classification algorithm (with merged classes) distinguished Urtica and Parietaria from
Brousonetia despite these pollen grains being morphologically similar."
Or
"It is interesting to note that the classification algorithm distinguished Urtica and Parietaria from Brousonetia with high
accuracy, despite these pollen grains being morphologically similar."
**Reply:** Yes, it is correct. And we appreciate the suggestion for improving clarity. The sentence now reads:
"It is interesting to note that the classification algorithm distinguished Urtica and Parietaria from Brousonetia with high
accuracy, despite these pollen grains are morphologically similar."

Fig. 2: The numbers and names are a bit small and blurry, would be good to make the characters a little larger if possible.
**Reply:** The figures were created in sufficient resolution, and we believe importing them into Word may have affected their
quality. We expect that in the published version, after typesetting, the original files will be used, so they won't be blurry. Since
the confusion matrices present 27 classes, increasing the font size is not feasible. Therefore, we suggest arranging the panels
of Figure 2 in a vertical orientation, which could result in a 100% increase in the size of the panel and thus improve the font
size as well.

(A)

[Figure]

(B)

[Figure]

Figure 2: Confusion matrices depicting pollen classification performance on test dataset measured in (A) "pollen mode" and
(B)
Line 226: what are the exact dates referred to here?
**Reply:** The dates for the indicated period, 3-7 May 2023, were added.
Line 235: Best to define PSLs in brackets for good measure as it is mentioned for the first time in this manuscript.
**Reply:** The "(Polystyrene Particles)" was added after PSLs when mentioned for the first time as suggested.
Line 241: At a glance, this sentence was a little confusing, I would correct it to something like: "Automatic detections of total
pollen, as well as Juglans, Morus and Ambrosia, have a statistically significant positive correlation with…"
**Reply:** Corrected as suggested.

Line 243: "for most pollen classes" or "for most of the pollen classes"
**Reply:** Corrected as suggested.

Line 245: Perhaps rephrase as, for example, "Pollen grains that occur simultaneously in the air had a clear tendency to be confused amongst each other, which was expected…"
**Reply:** We kept the original sentence here.

Line 261: "As demonstrated for the Rapid-E, this problem also exists for the Rapid-E+."
**Reply:** This sentence is changed following the suggestion from other participant of the public discussion, and section now reads:
"As a result, classification performance falls when a model trained on a reference dataset from one device is tested on a reference dataset from another one, which was demonstrated for Rapid-E (Matavulj et al., 2021). The same problem exists in Rapid-E+ (Fig. 4). The algorithm created on the training dataset collected with the Novi Sad device failed to identify the same reference pollen collected with both Osijek and FMI devices (average F1 score = 0.01 in both cases)"

Line 278: I would probably start a new sentence and replace the second i.e. before 'different timing…' with something else. This sentence is a bit confusing and long. Is it saying that since some pollen classes were comparable across devices, the differences observed across others shouldn't be due to doing lab work at different times and different methods of pollen exposure to the instrument? Or are you saying each lab followed the same procedures so it shouldn't be an issue?
**Reply:** This section is shortened and now reads:
"When analysing the results of the cleaning reference data for the same pollen measured with different devices, we noticed a significant difference for most pollen classes, except for Platanus, Salix and Betula. Different timing of the lab work and different methods of exposing the device to pollen cannot explain observed differences but it is rather attributed to differences in device sensitivity to the scattering and/or fluorescence signals."

Fig. 5 writing font too small and am unsure what I am looking at in 5D, can labels be added to the x, y and colour axes?
**Reply:** We have increased the font size used in Figure 5. Regarding panel D, it presents an image from the scattering light as
described in the sub-chapter 2.2: "In addition, the intensity of light, scattering from a 637 nm laser, is recorded as an image
using a 4x4 pixel detector". We have expanded the caption of Figure 5 to give more details.

[Figure]

Figure 5: Comparison of reference Betula pollen measurements in "pollen mode" on Novi Sad, Osijek and FMI Rapid-E+
devices after preprocessing: (A) average 447 nm laser perpendicular polarisation scatter, (B) average 447 nm laser parallel
polarisation scatter, (C) histogram of size distribution (D) average unitless intensity of 637 nm laser scattered light, recorded
as an image using a 4x4 pixel detector.

Fig. 6 again writing font too small.
**Reply:** We have increased the font size used in Figure 6.

[Figure]

**References**

Matavulj, P., Brdar, S., Racković, M., Sikoparija, B. and Athanasiadis, I. N.: Domain adaptation with unlabeled data for model transferability between airborne particle identifiers. in: Proceedings of the 17th International Conference on Machine Learning and Data Mining (MLDM 2021), New York, USA, https://doi.org/10.5281/zenodo.5574164, 2021.

Sikoparija, B., Marko, O., Panic, M., Jakovetic, D., and Radisic, P.: How to prepare a pollen calendar for forecasting daily pollen concentrations of Ambrosia, Betula and Poaceae?, Aerobiologia, 34, 203-217, https://doi.org/10.1007/s10453-018-9507-9, 2018.